# HSP70 Multi-Functionality in Cancer

**DOI:** 10.3390/cells9030587

**Published:** 2020-03-02

**Authors:** Zarema Albakova, Grigoriy A. Armeev, Leonid M. Kanevskiy, Elena I. Kovalenko, Alexander M. Sapozhnikov

**Affiliations:** 1Department of Biology, Lomonosov Moscow State University, 119192 Moscow, Russia; armeev@intbio.org (G.A.A.); amsap@mail.ru (A.M.S.); 2Department of Immunology, Shemyakin and Ovchinnikov Institute of Bioorganic Chemistry of the Russian Academy of Sciences, 117997 Moscow, Russia; leonid_kanewski@mail.ru (L.M.K.); lenkovalen@mail.ru (E.I.K.)

**Keywords:** HSP70, cancer, immunotherapy, apoptosis, authophagy, senescence, metastasis, exosomes

## Abstract

The 70-kDa heat shock proteins (HSP70s) are abundantly present in cancer, providing malignant cells selective advantage by suppressing multiple apoptotic pathways, regulating necrosis, bypassing cellular senescence program, interfering with tumor immunity, promoting angiogenesis and supporting metastasis. This direct involvement of HSP70 in most of the cancer hallmarks explains the phenomenon of cancer “addiction” to HSP70, tightly linking tumor survival and growth to the HSP70 expression. HSP70 operates in different states through its catalytic cycle, suggesting that it can multi-function in malignant cells in any of these states. Clinically, tumor cells intensively release HSP70 in extracellular microenvironment, resulting in diverse outcomes for patient survival. Given its clinical significance, small molecule inhibitors were developed to target different sites of the HSP70 machinery. Furthermore, several HSP70-based immunotherapy approaches were assessed in clinical trials. This review will explore different roles of HSP70 on cancer progression and emphasize the importance of understanding the flexibility of HSP70 nature for future development of anti-cancer therapies.

## 1. Introduction

The discovery of impact of heat on chromosome puffing patterns of *Drosophila* in 1962 by Ritossa and the identification of heat-inducible genes and proteins opened a new field of research on heat shock response [1,2,3,4]. Later it became clear that high expression of heat shock proteins (HSPs), and especially 70 kDa proteins (HSP70), protects cells against stress [4,5,6]. Further research revealed the chaperoning function of HSP70 being responsible for its ability to enhance cell survival through its ability to catalyze reassembly of damaged ribonucleoproteins [4,7]. Serving as a molecular chaperone, HSP70 mediates a wide range of house-keeping activities (reviewed in [8]). House-keeping and stress-related functions of HSP70 include de novo protein folding and refolding, prevention of protein aggregation, degradation of proteins, transport of proteins across membranes, assembly and disassembly of protein complexes.

The HSP70 family is highly conserved in evolution [9,10,11]. Multiple HSP70s present in both prokaryotes and eukaryotes. In humans, 13 HSP70 homologues are found in different compartments (cytosol, nucleus, lysosomes, ER and mitochondria), suggesting individual and organelle-specific biological roles (reviewed in [4]) [8]. Long lines of experimental evidence suggest a crucial role of HSP70 in cancer [12]. It is highly expressed in malignant tumors and typically serves as a biomarker for poor prognosis [13,14]. In this review, we describe the structure and cycle of HSP70 machinery, HSP70 routes of transport to the extracellular milieu, emerging research on the diversity of HSP70 functions in cancer in relationship to the currently established Hanahan and Weinberg model of the hallmarks of cancer [15]. We highlight the importance of understanding the flexibility of HSP70 machinery for efficient developments of anti-cancer therapeutics. 

## 2. The HSP70 Machinery

The central function of the HSP70 chaperones is that they do not work alone, but rather as machinery of HSP70 and (co)chaperones collaborating with each other [16]. To perform such collaborations, HSP70 can operate in different states during its functional cycle. Co-chaperones that are involved in HSP70 functional cycle form an internal HSP70 network. During its functional cycle and within its internal network HSP70 directly interacts with the client proteins to perform its chaperone function. Concomitantly, HSP70 can handover client proteins to other (co)chaperone machines for further folding or degradation and this will be further referred to as an external HSP70 network.

### 2.1. HSP70 Structure 

A full-length crystal structure of human HSP70 in either its free or closed conformation has not yet been obtained. Structures of its two major domains, namely N-terminal nucleotide-binding domain (NBD, ~45kDa), responsible for ATPase activity, and C-terminal substrate-binding domain (SBD, ~25kDa), required for peptide binding, have been determined independently of each other in free or bound states [17,18,19,20,21,22,23,24,25,26,27,28,29]. The two HSP70 domains are connected by the linker (13aa) (Figure 1A) [28]. HSP70 binds to exposed hydrophobic residues on unfolded proteins and unlike HSP90, HSP70 does not have specific clients for binding [30]. From the HSP70-structural point of view, further studies should be performed to fulfill the currently missing communication between its two domains.

### 2.2. HSP70 Functional Cycle

In 1995 Ha and McKay demonstrated that constitutive isoform of HSP70, Heat shock cognate 70 (HSC70), undergoes several conformational changes through its ATPase cycle (Figure 1) [31]. The binding of polypeptide to SBD is tightly regulated by the NBD nucleotide states either being ADP-, ATP- bound or nucleotide-free. In the ADP-bound state, SBD binds to polypeptides with high affinity and slow association and dissociation rates [8]. In ADP-bound state SBDα forms the lid over the SBDβ, resulting in a closed conformation of SBD (Figure 1C) [18,28,31]. ATP binding to NBD decreases the affinity of SBD for substrate, increasing association and dissociation rates by 100-fold and 1000-fold, respectively (Figure 1A) [8,31,32,33,34]. This mechanism of rapid association of peptide and its timely release allows HSP70 to prevent peptide aggregation and perform its folding function [8,35]. Therefore, ATP-ADP exchange is crucial for HSP70 polypeptide uptake and release.

### 2.3. The Internal HSP70 Network

HSP70 functional cycle is regulated by co-chaperone HSP40 and nucleotide exchange factors (NEFs) (Figure 1) [36,37]. HSP40 chaperones, also known as J-domain proteins (JDPs), play important roles in HSP70 machinery. J-domain of HSP40 is involved in the stimulation of ATPase activity of the HSP70-NBD domain, whereas HSP40 C-terminal peptide-binding domain 1 (CTD1) interacts with C-tail of HSP70 (Figure 1B) [16,38,39]. HSP40 binds to non-native clients and engages HSP70 in its low-affinity ATP state. Binding of HSP40 to HSP70 displaces the client to HSP70 C-tail, causing HSP40 dissociation from the HSP70-HSP40 complex and leaving the client bound to HSP70 in its ADP state [39]. Therefore, HSP40 delivers clients to HSP70 and stimulates ATPase domain, shifting HSP70 towards its high-affinity (ADP) state. 

Interestingly, there are much more HSP40 isoforms found in cells than HSP70s [16,40,41]. Moreover, much lower concentration of J-protein is required to that of HSP70 to stimulate client folding [42]. Increase in cellular level of J-protein without rising up HSP70 level results in reduced refolding of client proteins [16]. For some cellular functions full structure of J-protein is not required for its J-domain is sufficient to stimulate HSP70 ATPase activity [16]. In such cases, there will be no direct J-protein interaction with the client itself, while increase in just J-domain level in a specific compartment will direct HSP70 to particular clients at this site [16]. Several HSP40 family members were described in tumor growth, however, their role is still not clearly defined (reviewed in [43]). Hence, understanding the effects of diversity of HSP40 members on HSP70 and how HSP40s manipulate HSP70 functions can be advantageous for the development of HSP70 modulators targeting specific locations in cancer research. 

Contrary to HSP40, which triggers HSP70 transition to ADP-bound state, NEFs facilitate ADP dissociation from NBD, resulting in rebinding of ATP and substrate release (Figure 1D) [8]. Eukaryotic NEFs include Bcl2-associated anthanogen (BAG), HSP110, HspBP1, each using distinct mechanism for nucleotide exchange (reviewed in [44]). It is important to note that NEFs do not exchange nucleotides, but rather use their bulk mass to stabilize open conformation of HSP70 [44,45]. Binding of ATP displaces NEF from NBD for the next round of peptide binding [44]. Intriguingly, NEF function can be attenuated by the HSP70-interacting protein (Hip). Hip preferentially binds to ADP-bound state of HSP70-peptide complexes, thereby, slowing ADP dissociation from NBD [46]. Hip showed to promote protein ubiquitination and degradation [47]. Several antitumor compounds were described to successfully mimic Hip function [47,48]. Therefore, Hip is involved in prevention of protein aggregation by competing with NEF for proceeding to the next stage of the HSP70 cycle or transfer of proteins to degradation. 

**Figure 1 cells-09-00587-f001:**
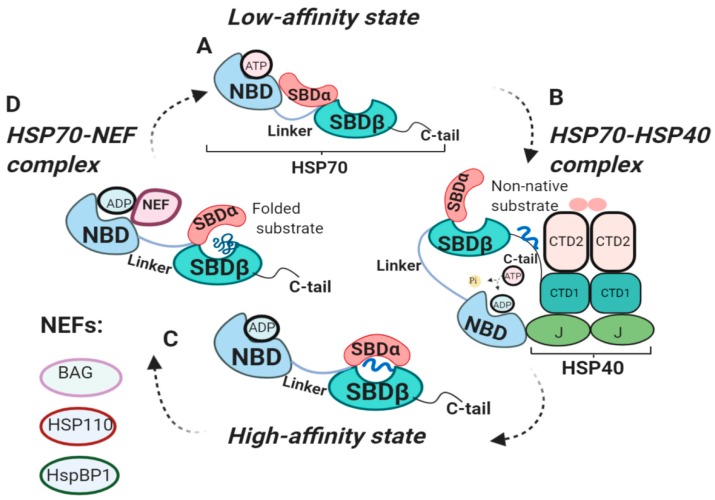
Functional cycle of HSP70 chaperones. (**A**) Structure of HSP70 in low-affinity (ATP) state. ATP binds to NBD, resulting in an open conformation of SBD, ready for client binding [8]. (**B**) HSP70-HSP40 complex. HSP40 presents non-native clients to HSP70. J-domain of HSP40 binds to HSP70-NBD stimulating its ATPase activity. Binding of HSP70 C-tail to HSP40-CTD1 displaces the client to HSP70, shifting HSP70 to ADP-bound state [16,38,39]. (**C**) High-affinity (ADP) state. ADP binds to NBD, SBDα forms the lid over SBDβ, locking substrate in SBD [18,28]. (**D**) HSP70-NEF complex. NEF displaces ADP from NBD, allowing ATP to bind NBD, shifting the HSP70 to low-affinity (ATP) state [8].NBD, nucleotide-binding domain; SBDα/β, substrate-binding domain; NEF, nucleotide exchange factor; CTD1/2, C-terminal peptide-binding domain of HSP40; J; J domain of HSP40.

### 2.4. The External HSP70 Network

Newly synthesized peptides will be either released from HSP70 for spontaneous folding, transferred further to HSP90 chaperone or will be degraded [8]. The interaction between HSP70 and HSP90 is mediated by HSP70-HSP90 organizing protein (HOP) (Figure 2A) [49,50,51]. HOP primarily consists of tetratrico-peptide repeat (TPR) domains, acting as a bridge between HSP70 and HSP90 for passing a client protein [49]. In cases when substrates spend too long time bound to HSP70 and cannot be refolded or handed over to HSP90, these substrates are targeted for degradation mediated by C-terminus of HSP70 interacting protein (CHIP) (Figure 2B) [52,53,54,55]. CHIP is an E3 ubiquitin ligase which works as a co-chaperone, competing with HOP for binding to HSP70 C-terminus and SBDα through its TPR-domain [56,57].

There is a growing interest in both academia and pharmaceutical industry in targeting HSP90 in cancer research [58]. HSP90 has a wide range of clients, some of which are oncogenic proteins, responsible for driving tumor transformation and progression [59,60]. Notably, HSP70-HSP90 multi-chaperone complex termed ‘epichaperome’ was found to be expressed in more than half of all tumors [61]. It is interesting to point out that this tight association in the epichaperome complex promotes tumor survival simultaneously making malignant cells more sensitive to HSP90 inhibitors [61]. 

## 3. HSP70 Transport to the Extracellular Milieu

The plasma membrane connects internal and external cellular signals [62]. In this regard, high expression of cytosolic HSP70 chaperones, lacking transmembrane sequence, was found on the surface of tumor cells, suggesting that there is a “cross-talk” between HSP70 and plasma membrane [62,63,64]. Multhoff and colleagues showed that HSP70 does not utilize classical endoplasmic reticulum-Golgi transport pathway for its membrane translocation [65]. Taking into account HSP70 binding abilities, authors deduced that HSP70 is transported to the plasma membrane by binding to other proteins [65]. Intriguingly, a recent report has demonstrated that co-chaperone HOP is required for HSP90–mediated deformation of the membrane and the fusion of multivesicular bodies to plasma membrane for the release of exosomes [66]. These findings suggest that HSP70 interaction with HSP90 via HOP can serve as indirect mechanism of HSP70 translocation to the surface of tumor cells. In light of the reported, HSP70 showed to be secreted from cells via exosomes [67,68,69,70]. Furthermore, Mambula and co-workers reported that HSP70 can be released via lysosomal endosomes [71]. Further research showed that high HSP70 levels resulted in accumulation of HSP70 in lysosomes leading to active HSP70 transport to the membrane and release of soluble HSP70 into the extracellular space [72]. Therefore, the endolysosomal route provides plasma membrane surface-bound, exosomal and soluble forms of HSP70 [73]. In addition, it was also suggested that HSP70 interaction with lipid vesicles containing phosphatidylserine (PS) or lipid raft component globoyltriaosylceramide (Gb3) allows integration of HSP70 into the plasma membrane [74,75,76,77]. Along this line, HSP70 also showed to bind 3′-sulfogalactolipids via its NBD domain [78]. Further research is needed to understand the process by which HSP70 is transported to the plasma membrane and released into the extracellular space.

## 4. HSP70 Receptors

Several receptors have been reported to interact with HSP70. These receptors include CD14, Toll-like receptors (TLR2 and TLR4), lectin-like oxidized low-density lipoprotein-1 (LOX-1), CD91, CD40 and receptor for advanced glycation endproducts (RAGE) [79,80,81,82,83,84,85,86]. Furthermore, natural killer (NK) cell receptors such as CD94/NKG2C, NKG2D as well as NKp30, NKp44, NKp46, NKp80 showed to be upregulated in the presence of HSP70-derived TKD peptide, suggesting involvement of these receptors in mediating the interaction of NK cells with surface-bound HSP70 [73,87,88,89]. 

Large proportions of HSP70 receptors were identified using recombinant HSP70 or HSP70-peptide complexes [79,80,81,82,83,84,85,89]. In most of the cases, recombinant endotoxin-free HSP70 or HSP70 purified using ATP-agarose column will only provide us with information on HSP70 in its low affinity state (Figure 1A), thus, missing possible potential interactions of HSP70 with receptors in its other conformations (Figure 1B–D). Conversely, HSP70-peptide complexes will provide us with information on HSP70 receptor binding in its high-affinity state (Figure 1C). Taking into account the ability of HSP70 to bind substrate in one period of time or bind substrate and interact with HOP (for handing over the substrate to HSP90) in another period of time or engage CHIP (for substrate degradation), suggests that state of HSP70 may affect its binding abilities, as otherwise all these interactions would happen simultaneously.

Therefore, in order to evaluate the spectrum of interactions of HSP70 with a variety of receptors and other effector molecules, we have performed molecular docking studies. Since a full-length crystal structure of human HSP70 protein has not yet been obtained, we have used the structures of its two major domains such as NBD and SBD in their bound and free forms. The structures of HSP70 domains and all proteins/receptors used for docking were obtained from the Protein Data Bank (PDB) (Appendix A) [17,18,19,20,21,22,23,24,25,26,27,28,29,90,91,92,93,94,95,96,97,98,99,100,101,102,103,104,105,106,107,108,109,110,111,112,113,114,115]. The molecules of interest were extracted from the models according to the identifier of the peptide chain (Appendix A). The linker regions of HSP70, membrane regions and other unnecessary regions of target molecules were excluded from binding region search. All the HSP70-target pairs were subjected to the protein-protein docking procedure using the ZDOCK software (6 degrees of freedom search of 1000 poses) [116].The results were ranked by energy using the ZRANK software, where the complexes with the lowest energy were selected as the most stable ones [117]. The obtained interaction energies are shown in Figure 3. Consequently, all energies were then used to average interacting molecules within the class and were translated to the Z score (Appendix A). 

Free-form NBD on average showed weak binding to potential targets than NBD in ATP- or ADP-bound forms (Figure 3; Appendix A). Interestingly, the opposite picture is seen for SBD: the free-form showed to have the lowest energy, representing the highest possibility for interaction. It can be concluded that HSP70 bound to peptide has a potential to bind different receptors and this interaction is mediated by its NBD. Since J domain of HSP40 is capable of binding to HSP70-NBD domain, it can be hypothesized that similar binding mechanism is used by HSP70 for binding to different receptors [16]. Taking into account that HSP70 does not have high specificity for client proteins, it can be hypothesized that unbound-HSP70 has potential to bind receptors and this is mediated by its SBD domain. Future molecular docking procedures can further assess the interactions of various receptors with HSP70 bound to either BAG3, HSP40, HOP-HSP90 complex and CHIP to provide further understanding of HSP70 binding potential in its different conformational states. It is important to note that the molecular docking does not allow to unambiguously detect interactions in all cases, but such predictions make it possible to narrow down the set of candidates for further computational and experimental interaction screening. Taking into account the complexity of HSP70 protein and its potential to bind different client proteins as well its ability to interact with (co)chaperone machines, molecular docking predictions may provide important insights into understanding of the possible HSP70 interactions. However, results obtained from molecular docking studies should be further verified by experimental data. 

## 5. Diverse Functions of HSP70 in the Hallmarks of Cancer

Being abundantly expressed in cancer, HSP70s have a wide range of activities. With a particular focus on a major stress-inducible HSP70 member, we will examine here HSP70 involvement in the hallmarks of cancer—a concept of multistep tumor development originally proposed by Hanahan and Weinberg (Figure 4) [15,118].

### 5.1. HSP70 and Tumor Immunity

Cancer progresses over years continuously shaping its interaction with an immune system. The role of HSP70 in that interaction is not yet completely understood making it one of the major endeavors in cancer immune biology. Immunogenic nature of HSP70 comes from its ability to bind antigenic peptides derived from tumors [119]. Blachere and colleagues showed that HSP70- peptide complex resulted in antigen-specific CD8+ T cell response [120]. This experiment has opened new perspectives for the use of HSP70 as an adjuvant [119]. 

HSP70 is highly expressed on the surface of tumor cells [65]. These HSP70-membrane positive tumor cells actively release HSP70 surface-positive exosomes leading to the stimulation of NK cells [67]. Multhoff and colleagues showed that surface-bound HSP70s are recognized by NK cells pre-stimulated with HSP70 protein or 14-mer HSP70 peptide (TKD) in the presence of interleukin-2 (IL-2) or IL-15 cytokines [88,121,122,123,124]. Notably, TKD peptide alone is not sufficient to stimulate NK cells, requiring the presence of IL-2 or IL-15 [87,88,121]. In 2015 phase II clinical trial was initiated to study the efficacy of autologous NK cells pre-activated with TKD and IL-2 following radiochemotherapy on patients with non-small cell lung carcinoma [124]. 

Considering results from the clinic, it is important to point out that there are two types of HSP70 circulating in the serum of cancer patients. One of the types is exosomal HSP70 released by viable tumor cells, whereas another is HSP70 released by dying cancer cells serving as damage-associated molecular patterns (DAMPs) [125,126,127].

Acting as a DAMP released by necrotic cells, HSP70 has great immunogenic potential for eliciting strong anti-tumor T cell response either being bound to tumor antigen or antigen-free [128,129]. However, long-term exposure of immune cells to free HSP70 following radiotherapy showed to induce immune tolerance and promote tumor growth [130]. These findings are consistent with report showing that low dose of HSP70-peptide complex is sufficient to stimulate antitumor immunity [120]. Translating this to the endogenous HSP70 released from tumor cells, it can be proposed that initial release of HSP70 acts as tumor suppressant, while HSP70 overload leads to tumor progression. 

HSP70-membrane positive phenotype (mHSP70) showed controversial results in predicting patients’ prognosis where positive clinical outcome was achieved for gastric and colon cancer, supporting the idea of NK targeting mHSP70, and poor prognosis in lower rectal and squamous cell carcinomas [131]. Authors deduced that differences in clinical outcomes can be explained by the differences in metastatic routes of these cancers [131]. Therefore, HSP70 release in extracellular microenvironment may lead to killing of mHSP70 tumor cells by NK cells or result in immune tolerance to tumor antigens. 

HSP70 showed to stimulate both innate and adaptive immune responses [132]. In this regard, extracellular HSP70 (eHSP70) activated T regulatory cells resulting in downregulation of interferon-γ (IFN-γ) and tumor necrosis factor-α (TNF-α) and upregulation of IL-10 and transforming growth factor-β (TGF-β) cytokines [133]. The eHSP70 interaction with antigen-presenting cells (APC) led to the stimulation of inflammatory cytokines such as TNF-α, IL-1β, IL-6 and IL-12 via activation of nuclear factor kappa B (NF-kB) [79,80,134,135]. Furthermore, HSP70 showed to negatively regulate critical component of innate immune response – the Nod-like receptor protein-3 (NLRP3) inflammasome [136]. Taking into account cytosolic location of NLRP3 inflammasome, its ability to sense DAMPs and its dysregulation in cancer, further studies should be performed to understand the effect of intracellular HSP70 on NLRP3 inflammasome in the context of cancer [137]. In light of the reported, other innate immunity molecule Tag7 showed to form a complex with HSP70 [138]. This Tag7-HSP70 complex interacts with Tumor necrosis factor receptor 1 (TNFR1) leading to permeabilization of lysosomal membrane and subsequent tumor cell death [73,138]. T lymphocytes also showed to utilize Tag7-HSP70 cytotoxic complex [139]. 

HSP70-positive exosomes showed to activate myeloid-derived suppressor cells (MDSCs) leading to IL-6 production [140]. Intriguingly, Bausero and co-workers (2004) demonstrated that IL-10 and IFN-γ cytokines lead to active release of HSP70 in exosomes by increasing intracellular level of HSP70 [141]. Since high levels of IL-10 and IFN-γ have been found in the serum of cancer patients, it can be hypothesized that these cytokines initially mark tumor cells for immune recognition in the form of HSP70-bound exosomes, while chronic activation of these cytokines and their subsequent effect on promoting HSP70 overexpression may lead to immune tolerance [142,143]. However, further studies should be performed for a better understanding of relationships between cytokines and HSP70 in shaping tumor immunity. 

The concept of HSP70-based vaccines was proposed utilizing the ability of HSP-peptide complexes to elicit cytotoxic T lymphocyte (CTL) response by cross-priming. In that context, HSP70-peptide complexes extracted from tumor were used to immunize the host, thus, delivering these complexes to antigen-presenting cells (APCs) for T cell priming through MHC class I pathway [144]. Notably, HSP70-peptide complexes isolated from fusion of tumor and dendritic cells (DCs) showed the most promising results [145]. Other HSP70-based immunotherapy approaches which have entered clinical trials will be discussed later in this review.

In light of that, one of the top 10 challenges described by Hegde and Chen (2020) in cancer immunotherapy is to determine dominant drivers of tumor immunity [146]. In this regard, HSP70 showed to be clinically important in different types of cancer and tumor immunology, however, further research is needed to determine its role in driving tumor immunity.

### 5.2. HSP70 and Tumor Resistance to Cell Death

Investigators exploring the role of HSPs on malignant cells proposed that tumor cells are “addicted to chaperones” [147]. Indeed, tumor cells survive under conditions of continuous stress such as hypoxia and nutrient starvation which make them susceptible to apoptosis [15]. HSP70, which is abundantly found in cancers, showed to suppress apoptosis, allowing cancer to progress [148]. Notably, decrease in the level of HSP70 showed to induce cancer cell death [149]. 

Evidently, HSP70 showed to affect both intrinsic and extrinsic apoptotic pathways (Figure 5). In the model of TNF-α induced apoptosis it was reported that the HSP70-CHIP complex promotes proteasomal degradation of apoptosis signal-regulating kinase 1 (ASK1) resulting in inhibition of c-Jun N-terminal kinase (JNK) and p38 (Figure 5A) [150]. JNK activity showed to be important for the release of cytochrome *c* from the mitochondria for the induction of intrinsic apoptotic pathway [151]. This JNK-dependent apoptosis involves p53 for the upregulation of Bcl-2-associated X protein (Bax) and BH3 interacting-domain death agonist (Bid) [152,153]. In addition, HSP70 inhibits apoptosis-inducing factor (AIF) required for DNA fragmentation [154]. In extrinsic pathway, HSP70 showed to block the formation of death-inducing signalling complex (DISC) through its interaction with TNF-related apoptosis-inducing ligand receptor 1 (TRAIL-R1) and TRAIL-R2 (Figure 5B) [155]. This may also result in an impaired cleavage of Bid by caspase-8 [156]. Therefore, HSP70 suppresses apoptosis by caspase-dependent mechanism via inhibition of JNK, caspase-independent mechanism via inhibition of AIF and through the interaction with death receptors.

Apart from apoptosis, HSP70 plays an important role in alternative forms of cell death including autophagy and necrosis. In contrast to normal cells, HSP70 was found to be present in the lysosomes of cancer cells [157]. Several studies have demonstrated that HSP70 stabilizes lysosomal membranes allowing cancer cells to escape cell death (Figure 5A) [157,158,159]. HSP70 inhibitor apoptozole showed to be effective for promoting apoptosis in cancer cells by triggering lysosomal membrane permeabilization and cathepsin release [160]. 

HSP70 protection from necrotic cell death is mediated through inhibition of JNK and does not require its chaperone activity [161]. As discussed earlier, HSP70 acts as DAMP and can initially elicit strong immunogenic response which with chronic exposure leads to immune tolerance. Therefore, it can be concluded that HSP70 plays dual role in necrosis of malignant cells. On the one hand, following necrosis HSP70 may induce anti-tumor response. On the other hand, HSP70 protects cancer cells from the necrotic cell death, thus, allowing for tumor progression which appears to be the case with long-term anti-cancer treatment accompanied by the sustained release of necrotic signals, leading to tumor growth and resistance to therapy [162]. 

**Figure 5 cells-09-00587-f005:**
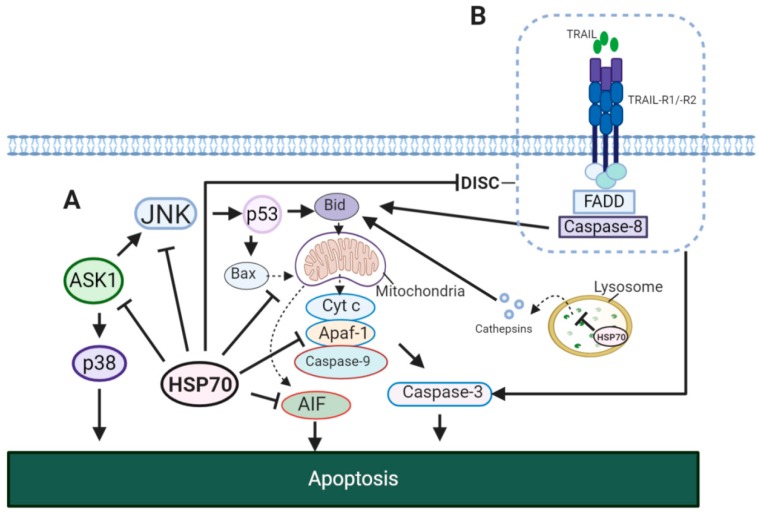
HSP70 in apoptotic signaling pathways. (**A**) HSP70 in intrinsic apoptosis. In TNF-α induced apoptosis, HSP70 inhibits JNK and p38, thus, preventing the translocation of Bax and Bid to the mitochondria [150,156,163]. This blocks the release of cytochrome *c* to the cytosol and its formation of apoptosome complex with Apaf-1 and caspase-9. HSP70 may also disrupt the formation of apoptosome by binding to Apaf-1, preventing its association with procaspase-9 [164]. In addition, HSP70 stabilizes lysosomal membrane, inhibiting cathepsin release to the cytosol for the activation of intrinsic apoptotic pathway. (**B**) HSP70 in extrinsic apoptosis. HSP70 interacts with TRAIL-R1 and TRAIL-R2 and prevents the formation of DISC complex required for the initiation of extrinsic apoptosis. JNK, c-Jun N-terminal kinase; Cyt C, cytochrome *c*; Bax, Bcl-2-associated X protein; Bid, BH3 interacting-domain death agonist; p38, p38 mitogen-activated protein kinase; Apaf-1, apoptotic protease-activating factor 1; ASK, apoptosis signal-regulating kinase 1; TRAIL, TNF-related apoptosis-inducing ligand; TRAIL-R1/R2, TNF-related apoptosis-inducing ligand receptor 1/2; FADD, FAS-associated death domain (FADD).

### 5.3. HSP70 and Senescence Program

High expression of HSP70 is critical for the survival and growth of cancer cells as HSP70 depletion in cancer showed to activate oncogene-induced senescence program. HSP70 suppresses senescence by regulating p53 and cell cycle kinase Cdc2 [165]. Furthermore, depletion of HSP70 triggers the activation of p53-dependent or ERK-dependent senescence pathways depending on the expression of different oncogenes. For example, cells that express PIK3CA oncogene led to p53-dependent senescence, while cells with Ras oncogene expression activated ERK-dependent senescence [166]. Along this line, HSP70 downregulation showed to inhibit the transformation of Her2-expressing non-malignant mammary epithelial cells (MCF10A) leading to cellular senescence by upregulation of p21 and depletion of survivin [167]. Interestingly, results from another Her2 –positive cell line SK-BR-3 showed that p21 and survivin pathways function either in p53-dependent or p53-independent manner, respectively [167]. In addition, nucleotide-exchange factor BAG3 also showed to play a role in regulation of survivin and p21 in several cancer lines, suggesting potential role of HSP70-BAG3 complex in suppression of senescence [168]. 

It has been shown that downregulation of heat shock transcription factor 1 (HSF1) inhibits tumor growth and promotes cellular senescence by regulating HSP70 [169]. However, recent report showed that HSF1 can independently induce cellular senescence via p53-p21 pathway [170]. 

### 5.4. HSP70 in Sustained Proliferation

Recent reports have demonstrated that HSP70 in complex with HSP40 co-chaperone inactivates tumor suppressor protein p53 by interacting with its DNA binding domain, leading to p53 unfolded state [171,172]. These findings shed the light on why the depletion of HSP70 resulted in upregulation of p53 target molecule p21 and subsequent activation of senescence program discussed earlier in this review [167,171]. In contrast, transfer of p53 from HSP70 to HSP90 via HOP showed to stabilize p53 in its native conformation [171,172]. Therefore, HSP70-HSP40 and HSP70-HOP-HSP90 interplay is important in maintaining p53 conformational equilibrium [171].

Along this line, HSP70 showed to stabilize aminoacyl-transfer RNA synthetase-interacting multifunctional protein 2 (AIMP2) lacking exon 2 (AIMP2-DX2), the splicing variant of AIMP2. AIMP2 acts as pro-apoptotic factor positively regulating p53 in contrast to its splicing variant AIMP-DX2 which expression showed to correlate with aggressive tumor phenotype and chemoresistance [23,173]. AIMP2-DX2 escapes the Siah1- dependent ubiquitination by binding to HSP70 SBD domain [23]. Taken together, these data provide a good example on how HSP70 can exert its actions by manipulating the conformational state of its client proteins or protecting them from further degradation. Therefore, further studies are required for identifying other cancer-related client proteins of HSP70 machinery.

### 5.5. HSP70 in Angiogenesis

HSP70 and HSP90 showed to stabilize the accumulation of hypoxia-inducible factor-1α (HIF-1α), one of the two subunits of hypoxia-inducible factor-1(HIF-1) [174]. Importantly, HIF-1 which is responsible for sensing low level of oxygen showed to be involved in tumor angiogenesis, invasion and metastasis [174,175,176]. It has been reported that HSP70 binds and stabilizes HIF-1α in cancer cell lines [177]. Crucially, HIF-1 showed to be regulated by HSF1 via HuR, an RNA-binding protein overexpressed in cancer [178,179]. The effects of HSF1 on HIF-1 via HuR showed to be HSP70-independent, suggesting that HSP70 and HSF1 may affect HIF-1 through distinct mechanisms [178]. Kim and colleagues showed that extracellular HSP70 can bind to the surface of endothelial cells and induce angiogenesis by ERK-dependent mechanism [180]. In addition, it has been reported that HSP70 enhances IL-5-induced angiogenesis via endothelial nitric oxide synthase (eNOS) pathway, suggesting an inflammatory role of HSP70 in promoting angiogenesis [181].

### 5.6. HSP70 in Metastasis

High expression of HSP70 was shown to be associated with metastases to the lymph nodes in the breast cancer models [182]. Further research has found that inactivation of HSP70 showed to reduce tumor invasiveness and metastatic potential of breast, cervical and bladder cancer cell lines [183]. High HSP70 expression protects tumor cells against anoikis and amorphosis, forms of cell death that occur when cell loses its contact with extracellular matrix (ECM) [184,185]. In this regard, molecules involved in anoikis and amorphosis such as focal adhesion kinase (FAK) and Akt showed to be affected by HSP70 [186,187,188]. 

Epithelial-mesenchymal transition (EMT) is a complex process in which cancer cell acquires migratory and metastatic potential. Li and colleagues demonstrated that HSP70-peptide complexes (HSP70-PC) extracted from hepatocarcinoma samples induce EMT in human hepatoma (Huh-7) cell line acting through p38 MAPK signalling pathway [189]. In contrast, recent studies using small interfering RNA (siRNA) and short hairpin RNA (shRNA) have showed that *HSP70* inactivation promotes cell transition to mesenchymal phenotype [185,190]. HSP70 downregulation destabilizes E-cadherin-catenin complexes allowing migration of tumor cells [185]. In addition, several reports have demonstrated that HSP70 inhibits TGF-β-induced EMT by decreasing Smad2 phosphorylation [191,192,193]. Furthermore, it has been shown that HSP70 prevents high glucose-induced EMT of peritoneal mesothelial cells by inhibiting Smad3 and Smad4 phosphorylation and reactive oxygen species (ROS) production [194]. These findings suggest dual role of HSP70 in EMT, where HSP70-peptide complexes support transition of tumor cells towards acquiring mesenchymal phenotype. Intriguingly, HSP70 (co)chaperones also showed to be involved in tumor metastasis. One of the key EMT regulators Met is downregulated by C-terminus of HSP70 interacting protein (CHIP) [187,195]. High expression of HSP70’s co-chaperone HOP has been found in ovarian cancer and glioblastoma cells lines [196,197]. HOP inactivation has been reported to inhibit pseudopodia formation and migration of breast cancer cells [198]. Furthermore, HOP knockdown downregulates matrix metalloproteinase-2 (MMP-2), resulting in reduced invasion of pancreatic cancer cells [199]. Another HSP70’s co-chaperone BAG3 enables MMP-2 to promote invasion of ovarian cancer cells [200]. Extracellular HSP70 induces the release of MMP-9, a processing enzyme for cleavage of CD44 receptor involved in adhesion and migration [201,202]. In addition, HSP70 and HSP90 proteins showed to stabilize Wiskott-Aldridge syndrome family member 3 (WASF3), protein required for tumor migration, invasion and metastasis [203,204,205].

Clinically, circulating tumor cells (CTCs) that underwent EMT serve as a prognostic marker of a metastatic spread and can be isolated from patient’s blood using epithelial cell adhesion molecule (EpCAM) as a cell surface marker [206]. Interestingly, in contrast to EpCAM, the EMT does not downregulate HSP70 expression as higher membrane-bound HSP70 (mHSP70) expression has been found on metastatic tumors compared to its primary counterparts [131,206,207,208,209]. In this regard, Stangl and colleagues used HSP70-TKD peptide to develop monoclonal antibody directed against mHSP70 for specific targeting mHSP70-positive tumors by NK cells and efficient isolation of CTCs [206,210]. Taken together, on the one hand, high surface expression of HSP70 is found in metastatic tumors and mHSP70 may serve as a biomarker for assessing the metastatic spread. On the other hand, HSP70 plays dual role in EMT. The possible explanation can be that HSP70 works in a network where interaction of HSP70 either with BAG3 or HOP has a positive effect on tumor progression and this is consistent with the idea of HSP70-PC to promote EMT as HSP70 either bound to BAG3 or HOP may still carry a client protein. Conversely, interaction with CHIP suppresses EMT. Therefore, it can be hypothesized that different HSP70 roles in metastasis are specified through HSP70 interactions with its internal and external network of its (co)chaperones. However, further research is required in elucidating the role of HSP70 and its (co)chaperones in metastatic process. 

## 6. HSP70 Therapies Targeting Cancer

### 6.1. Small Molecule Inhibitors of HSP70 Cycle

Attempts were made to develop adenosine-derived HSP70 inhibitors using structure-based design approach [211,212]. Non-nucleotide ligands mimicking HSP70-NBD in ADP-bound state were proposed as well, however, from drug discovery perspective, it remains challenging to develop compounds targeting nucleotide-binding domain [19]. HSP70 small molecule inhibitors targeting cancer are summarized in Table 1. Interesting results were gained by Li and colleagues with JG-98 inhibitor. JG-98 disrupts HSP70-BAG3 complex, thereby, freezing HSP70 in its ADP state [213,214]. Similar effects were reported with YM-1, an inhibitor that targets HSP70-NEF complex, indicating that freezing of HSP70 in ADP-bound state can be the most promising strategy for the future developments of anti-cancer drugs [168]. In such state of tight association of HSP70 with peptide, this complex will be ubiquitinated by CHIP and further targeted for proteasomal degradation [47,215].

ATPase inhibitors proved to be effective in vitro and in vivo as one molecule can target multiple HSP70 isoforms and this strategy might be beneficial for cancer types with overexpression of multiple HSP70 isoforms [216,217,218]. However, “one-size-fits-all” approach may not work for distinct HSP70 family members involved in specific cancer type, in such cases it is important to consider isoform-specific inhibitors for future developments. Therefore, it is critical to gain better understanding of HSP70 functional cycle, including conformational changes and interaction between its two domains, the roles (co)chaperones play in HSP70 machinery as well as roles of HSP70 isoforms in different cancer types for future design of efficient HSP70 inhibitors for cancer therapy.

**Table 1 cells-09-00587-t001:** Small molecules targeting HSP70 functional cycle in cancer.

HSP70-Targeting Molecules		Effects	Refs.
**SBD-targeting inhibitors**	PES (Pifithrin-µ)	Suppressed tumor development in a mouse model of *Myc*-lymphoma.	[219]
	ADD70	Increased sensitivity of colon cancer and melanoma cells to apoptosis; showed antimetastatic effects in vivo.	[220]
	Acridizinium derivative 1	Induced apoptosis in HeLa cells.	[221]
**NBD-targeting inhibitors**	Synthetic peptide P17	Decreased melanoma growth in vivo.	[222]
	JG-98 inhibitor (MKT-077 analog)	Decreased tumor growth in MCF7 xenograft model.	[213]
	VER-155008	Inhibited proliferation of human breast and colon cancer cell lines.	[217]
	YK-5	Induced apoptosis and degradation of HSP70/HSP90 client proteins (HER2,Raf-1, Akt kinases) in breast cancer cells.	[223]
	Apoptozole	Induced apoptosis in ovarian, colon and lung cancer cell lines.	[224,225]
**HSP70-HSP40 complex inhibitors**	MAL3-101	Inhibited proliferation of multiple myeloma cells derived from patients.	[226]
	DMT3132 (MAL3-101 analog)	Reduced proliferation in breast cancer cells.	[227,228]
	Myricetin	Inhibited tumor growth in pancreatic cancer.	[229]
**HSP70-NEF complex inhibitors**	YM-1	Inhibited tumor growth in mammary and melanoma xenograft models.	[168]
**Allosteric HSP70 inhibitors**	HS-72	Inhibited growth in HER2-positive breast cancel model.	[230]

### 6.2. Other HSP70 Inhibitors

Triptolide which was originally isolated from *Tripterygium wilfordii* and its water-soluble pro-drug derivative minnelide showed to inhibit HSP70 expression in pancreatic cancer, neuroblastoma, mesothelioma, osteosarcoma and non-small cell lung carcinoma by inducing microRNA miR-142-3p binding to 3′UTR and downregulation of *HSP70* mRNA level [231,232,233,234,235,236,237]. Notably, minnelide has entered phase II clinical trial for the treatment of refractory pancreatic cancer [238]. Quercetin and demethoxycurcumin also showed to inhibit HSP70 in breast and prostate cancer cells, respectively [239,240,241].

### 6.3. HSP70-Based Therapies in Cancer Clinical Trials

There is an enormous challenge in developing safe and effective drug that can pass all phases of clinical trials to further be accessible for the treatment of cancer patients. In this regard, HSP70 showed to be effective in vitro and in vivo, however, the success of HSP70-based therapies in cancer can only be verified by assessing their efficacy and safety in clinical trials (Table 2). In that regard it is encouraging that HSP70-based therapies showed to be well tolerated among patients [124,242,243,244,245,246,247]. However, since most of these clinical trials, except phase II trial of Specht and colleagues, were conducted in 2003-2015 years and have not been progressed through drug development timeline and further pharmacovigilance, assessment of the efficacy and safety of HSP70-based therapies for the treatment of cancer patients still remains a major challenge.

## 7. Conclusions and Perspectives

HSP70 is a powerful chaperone highly expressed in different types of cancers and showed to associate with resistance to chemotherapy and poor prognosis for patients. HSP70 multi-functional role is specified through its interaction with co-chaperones HSP40 and nucleotide-exchange factors and the collaboration with another powerful chaperone network - HSP90. Multi-faced appearance of HSP70 is represented by its ability to change its conformations through its functional cycle, allowing HSP70 to operate in different states, the consequences of which in the context of cancer are not yet fully understood. Its ability to suppress apoptosis, senescence, immune responses and promote angiogenesis and metastasis makes it an attractive target for anti-cancer therapies. Attempts to develop small molecule inhibitors targeting its two major domains in different states showed promising results, however, it is not yet clear how it can be translated into clinics, taking into account ubiquitous expression of HSP70 and presence of different isoforms. Even though considerable progress has been made in elucidating functions of HSP70 in cancer, there is still a lot to be understood. For example, considering high expression of HSP40 in cancer and its tight association with HSP70, the roles of different HSP40 family members play in cancer are not yet identified. The role of extracellular HSP70 is also not clear. It also will be important to understand how different molecular functions of HSP70 relate to the clinical outcome. Further elucidating of HSP70 functions in the cancer hallmarks will allow for developing an efficient HSP70-based anti-cancer therapy.

## Figures and Tables

**Figure 2 cells-09-00587-f002:**
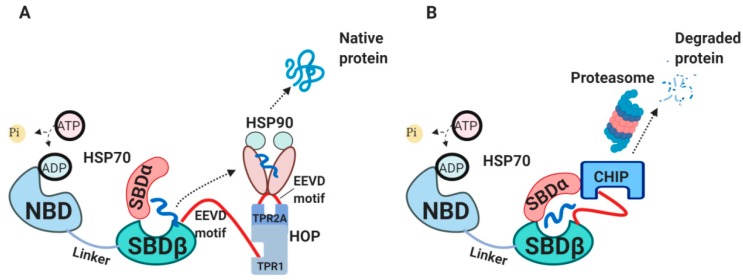
The external HSP70 network. (**A**) HSP70-HSP90-HOP complex. HOP acting as an adaptor molecule mediates the association between HSP70 and HSP90. HSP70 C-terminal EEVD motif binds TPR1 on HOP, whereas HSP90 C-terminal EEVD binds TPR2A [49]. This association via HOP allows handing over the substrate from HSP70 to HSP90 for further folding. (**B**) HSP70-CHIP complexes. CHIP binds to C-terminus of HSP70-substrate complex as well as to SBDα lid through its TPR domain, ubiquitylates HSP70-bound peptides and targets them for proteasomal degradation [55,57]. HOP, HSP70/HSP90-organizing protein; TPR, tetratrico-peptide repeats; CHIP, C-terminus of HSP70 interacting protein.

**Figure 3 cells-09-00587-f003:**
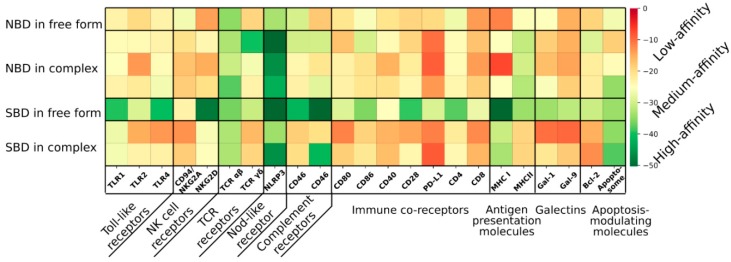
Heatmap of top docking predictions of various targets with HSP70 domains in bound and free forms. Color values correspond to binding energy assessed with the ZRANK scoring function (lower energies correspond to a higher probability of a complex formation).NBD, Nucleotide-binding domain; SBD, Substrate-binding domain.

**Figure 4 cells-09-00587-f004:**
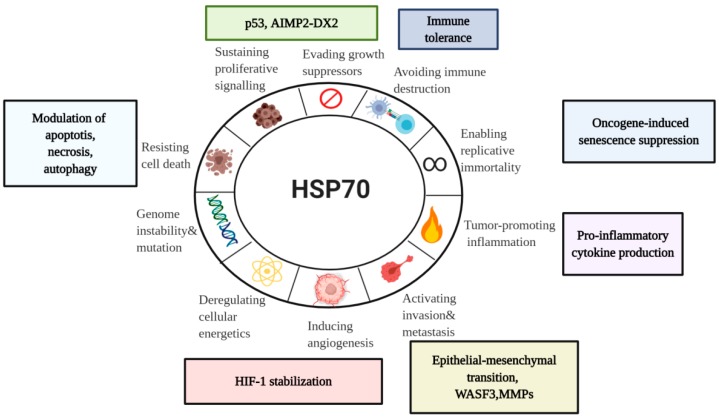
HSP70 in the Hallmarks of Cancer. Taking into account emerging role of HSP70 in cancer, it is important to assess its association with the currently established Hanahan and Weinberg model of the Hallmarks of Cancer [15]. Even though there are a lot of unanswered questions concerning HSP70 functions in tumor progression, current model can serve for the better understanding of diverse HSP70 roles in cancer for future development of anti-cancer therapeutics.WASF3, Wiskott-Aldridge syndrome family member 3; MMP, Matrix metalloproteinase; HIF-1, hypoxia-inducible factor-1; AIMP2-DX2,Aminoacyl-transfer RNA synthetase-interacting multifunctional protein 2 lacking exon 2

**Table 2 cells-09-00587-t002:** HSP70-based therapies in cancer clinical trials and case studies.

HSP70-Based Therapies	Condition	Clinical Trial Phase/Case Study	Refs.
Autologous HSP70-peptide complex in combination with imatinib mesylate	Chronic myeloid leukemia	Phase I	[247]
Autologous HSP70-peptide complex (AG-858) in combination with Gleevec	Chronic Myelogenous Leukemia	Phase II	[248]
Autologous peripheral blood mononuclear cells (PBMC) pre-activated with TKD peptide/IL-2	Colon carcinoma	Case study	[242]
Autologous NK cells pre-activated with TKD and IL-2 following radiochemotherapy	Advanced colorectal carcinoma and non- small cell lung carcinoma (NSCLC)	Phase I; Phase II	[124,243]
DNA vaccine expressing HPV16 E7 mutant form and HSP70	Cervical intraepithelial neoplasia	Phase I	[244]
Dendritic cells transfected with HSP70 mRNA	Hepatocellular carcinoma	Phase I	[245]
Autologous NK cells pre-treated with TKD/IL-2 in combination with radiochemotherapy and nivolumab (PD-1 antibody)	NSCLC stage IIIb	Case study	[246]
Intratumoral injection of recombinant oncolytic type 2 adenovirus overexpressing HSP70 (H103)	Advanced solid tumors	Phase I	[249]

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
