# Peer review of "HSP70 Multi-Functionality in Cancer"

_cells, 2020, doi:10.3390/cells9030587_

Round 1

Reviewer 1 Report

The manuscript deals with the role of Hsp70 in cancer. So much has been done looking at the role of the Hsp70 inhibitor triptolide and its derivative and the authors do not discuss  it. A comprehensive literature review is needed to enhance the quality of the manuscript.

Author Response

Response to Reviewer 1 Comments

Point 1: The manuscript deals with the role of Hsp70 in cancer. So much has been done looking at the role of the Hsp70 inhibitor triptolide and its derivative and the authors do not discuss  it.

Response 1:  Thank you for your comments. We have added the information on the role of triptolide and its derivative minnelide  (page 15, line 482)

Point 2: A comprehensive literature review is needed to enhance the quality of the manuscript

Response 2:  We have performed an extensive literature review in order to explore HSP70 as a therapeutic target in cancer. In this regard, we have added several HSP70-SBD inhibitors (page 14; Table 1), allosteric HSP70 inhibitors (pages 14-15, Table 1), we have also added Quercetin and demethoxycurcumin  to the group of “Other HSP70 inhibitors” (page 15, line 487) as well as HSP70-based therapies in clinical trials (page 15, line 490).

Reviewer 2 Report

The review concerns the role of Hsp70 in cancer. Overall it overviews massive data, and has a lot of accurate references. However, I have some concerns regarding the structure of the manuscript. At the moment, in the PubMed there are almost 400 reviews with keywords Hsp70 + cancer,  indicating the size of the field, and therefore a new review must have something unique. Unless it is a 100-page review, there is no possibility to reasonably overview the entire field. Accordingly, the current review is somewhat shallow. It gives a lot of facts in a very dense manner, which are not easy to understand. In order for readers to benefit from the review, it should be focused on some aspect of the field, e.g. immunity or maybe cell resistance, and should be less dense. 

Another problem, a good review usually is not just a description of a collection of facts, but rather places them in a model, so that a reader will be able to have in mind a general picture of the field. This review does not provide such a model.

Finally, the manuscript has a description of the computational work to find receptors for Hsp70. The table with interaction energies is difficult to understand. Why not just giving the manes of the proteins? In addition, it is not clear how meaningful all these data without wet-lab validation. 

Author Response

Response to Reviewer 2 Comments

Point 1: The review concerns the role of Hsp70 in cancer. Overall it overviews massive data, and has a lot of accurate references. However, I have some concerns regarding the structure of the manuscript. At the moment, in the PubMed there are almost 400 reviews with keywords Hsp70 + cancer,  indicating the size of the field, and therefore a new review must have something unique. Unless it is a 100-page review, there is no possibility to reasonably overview the entire field. Accordingly, the current review is somewhat shallow. It gives a lot of facts in a very dense manner, which are not easy to understand. In order for readers to benefit from the review, it should be focused on some aspect of the field, e.g. immunity or maybe cell resistance, and should be less dense. 

Response 1: Thank you for the comments. We truly appreciate them. The idea of the review was to provide a reader with a whole picture of how complex HSP70 machinery including its structure, functional cycle, its ability to work in a networks and how all these networks  function in cancer, covering all up-to-date information concerning HSP70-cancer related functions. Using molecular docking predictions, we also provide a reader with an idea on how flexible HSP70 is in its binding abilities. We re-structured all review to make it easier to understand for the reader.

Point 2: Another problem, a good review usually is not just a description of a collection of facts, but rather places them in a model, so that a reader will be able to have in mind a general picture of the field. This review does not provide such a model.

Response 2: We have created such model. We have divided HSP70 machinery on internal and external networks ( page 2, line 51; line 76; page 4, line 125).  We added information on receptors (page 6; line 176-181). We further placed HSP70 cancer-related functions in the currently established model of the “Hallmarks and Cancer” developed by Hanahan and Weinberg ( page 7, line 233). In this regard, we draw a figure where we present diverse functions of HSP70 in their association to the hallmarks of cancer to provide reader with the general picture of the field and the latest research on HSP70  (Figure 4; page 8; line 235). We added a section on effects of HSP70 on sustained proliferation as it is also one of the hallmarks (page 12, line 381). We divided section “HSP70 in angiogenesis and metastasis” into two sections such as “HSP70 in angiogenesis” and “HSP70 in metastasis” to provide more in-depth look at role of HSP70 in EMT (page 13, line 419). As general picture of the field should also provide some clinical experience, we have added chapter on clinical trials (page 15, line 490).  All changes we highlighted using “Track changes” function.

Point 3: Finally, the manuscript has a description of the computational work to find receptors for Hsp70. The table with interaction energies is difficult to understand. Why not just giving the manes of the proteins? In addition, it is not clear how meaningful all these data without wet-lab validation

Response 3: We have simplified the figure by deleting all energies, adding annotations – “low-affinity”, “medium-affinity”, “high-affinity”, deleting PDB codes (Figure 3, page 7). Adding names of proteins in each row may confuse the reader, as each receptor corresponds to two forms of NBD domain (NBD-bound, NBD-free) and two forms of SBD (SBD –bound, SBD-free). There was already a sentence on how computational work is related to experimental data  - “such predictions make it possible to narrow down the set of candidates for further computational and experimental interaction screening” (page 7, line 217) and that “results obtained from molecular docking should be further verified by experimental data” ( page 7; line 223). We further added- “Taking into account the complexity of HSP70 protein and its potential to bind different client proteins as well its ability to interact with (co)chaperone machines, molecular docking predictions may provide important insights into understanding of the possible HSP70 interactions” (Page 7, line 219).  We also added the future proposal for molecular docking investigators  - “Future molecular docking procedures can further assess the interactions of various receptors with HSP70 bound to either BAG3, HSP40, HOP-HSP90 complex and CHIP to provide further understanding of HSP70 binding potential in its different conformational states” (page7, line 214).

Reviewer 3 Report

This is a comprehensive work on the role of HSP70s and their co-chaperones in cancer. The authors nicely discriminate between the different roles of HSPs in the different subcellular and extracellular compartments: cytosolic, membrane-bound and extracellular HSPs.

Also different approaches to use HSPs as potential targets for cancer therapies have been included into the review. 

The review alos indicates the open questions which are yet not solved with respect to HSPs in cancer.

There are only very few comments which might further improve the Ms.

HSP70 receptors: in the list of NK receptors which have the capacity to interact with HSP70 the receptor CD94/NKG2C should be included into the text.

It would be also good to add a chapter which specualtes about the transport mechanisms of Hsp70 from the cytoslo to the emembraen and to the extracellular space.

With respect to the stimulatory capacity of extracellular HSPs it should be mentioned that Hsp70 alone is not sufficient to stimulate NK cell activity. Only in the context with pro-inflammator cytokines such as IL-2 IL-15 Hsp70 is able to activate NK cells. This might alos be an explanation why Hsp70 alone (or exosoaml Hsp70) is not stimualing NK cells.

Author Response

Response to Reviewer 3 Comments

Point 1: This is a comprehensive work on the role of HSP70s and their co-chaperones in cancer. The authors nicely discriminate between the different roles of HSPs in the different subcellular and extracellular compartments: cytosolic, membrane-bound and extracellular HSPs.

Also different approaches to use HSPs as potential targets for cancer therapies have been included into the review. 

The review alos indicates the open questions which are yet not solved with respect to HSPs in cancer.

Response 1: Thank you for the comments.

Point 2: There are only very few comments which might further improve the Ms.

HSP70 receptors: in the list of NK receptors which have the capacity to interact with HSP70 the receptor CD94/NKG2C should be included into the text.

.

Response 2: We have added information on interaction of HSP70 with CD94/NKG2C and other NK receptors (page 6; lines 178-181).

Point 3: It would be also good to add a chapter which specualtes about the transport mechanisms of Hsp70 from the cytoslo to the emembraen and to the extracellular space.

Response 3: We have added the chapter on HSP70 transport to the extracellular membrane and space (page 5; lines 154-174).

Point 4: With respect to the stimulatory capacity of extracellular HSPs it should be mentioned that Hsp70 alone is not sufficient to stimulate NK cell activity. Only in the context with pro-inflammator cytokines such as IL-2 IL-15 Hsp70 is able to activate NK cells. This might alos be an explanation why Hsp70 alone (or exosoaml Hsp70) is not stimualing NK cells.

Response 4: We added information on HSP70 protein and HSP70-derived peptide TKD to stimulate NK cells in presence of IL-2 or IL-15 (page 8, lines 254-259).

Round 2

Reviewer 1 Report

The authors have addressed the concerns raised.